

# Optimization of the cytogenetic protocol for *Pangasianodon hypophthalmus* (Sauvage, 1878) and *Clarias gariepinus* (Burchell, 1822)

Victor T. Okomoda[1], Ivan C.C. Koh[2], Anuar Hassan[2], Thumronk Amornsakun[3], Julia H.Z. Moh[4] and Sheriff Md Shahreza[2,4]

[1] Department of Fisheries and Aquaculture, University of Agriculture Makurdi, Makurdi, Benue, Nigeria
[2] Aquaculture and Fisheries, Universiti Malaysia Terengganu, Kuala Nerus, Terengganu, Malaysia
[3] Technology and Industries, Prince of Songkla University, Mucang, Puttani, Songkla, Thailand
[4] Institute of Tropical Aquaculture (AKUATROP), Universiti Malaysia Terengganu, Kuala Nerus, Terengganu, Malaysia

Corresponding authors
Victor T. Okomoda,
okomodavictor@yahoo.com
Sheriff Md Shahreza,
shahreza@umt.edu.my

## ABSTRACT

To obtain well spread chromosomes, the cytogenetic protocol for *Pangasianodon hypophthalmus* and *Clarias gariepinus* were optimized. This includes, the colchicine concentration (0.01%, 0.025%, 0.05%)/exposure duration (1, 3, and 5 h), hypotonic solution (distilled water or 0.075M KCl solution)/exposure duration (30 min, 1, and 2 h), the time of cell suspension preparation (at hypotonic treatment or before slide preparation) and chromosome aging period (0, 3, and 7 days in Carnoy's fixative). In addition, the type (i.e., fin, gill or kidney) and the amount of tissue (10, 50, 100 or 150 mg) were also investigated. Regardless of the species, the result obtained showed that well-spread chromosomes could be obtained using the following optimized protocol: Juveniles are injected with 0.05% colchicine (at one ml kg$^{-1}$) and allowed to swim for 3 h. Then, 50 mg of gill tissue is made into cell suspension in 0.075M KCl for 1 h. The cell suspension is treated in Carnoy's fixative (changed three times at 20 min interval) and then aged for 3 days. Finally, chromosome slides are made and stained with 10% Giemsa for 1 h.

## INTRODUCTION

The first level of genome analysis of any organism involves karyotyping of mitotic chromosomes to determine the genome organization at the cytological level (*Gorman, 1973*). It is an essential tool in providing basic information on breeding programs such as inter-specific/intergeneric hybridization (*Crego-Prieto et al., 2013*), polyploidy induction (*Christopher, Murugesan & Sukumaran, 2010*; *Gilna, Kuzma & Otts, 2014*; *Thresher et al., 2014*), and genetic improvement of commercial exploited or novel fish stocks targeted toward commercialization (*Zhu et al., 2012*). An effective method of chromosome preparation is essential for cytogenetic research (*Shao et al., 2010*). Several techniques have been optimized to obtain well spread mitotic chromosomes in fish and this may differ for

different species (e.g., as observed by *Karami et al., 2015*). These are usually in terms of chemical types, concentration, and duration of exposure. Firstly, the cells spindle fiber is arrested at metaphase by inoculation with a spindle poison (*Rieder & Palazzo, 1992*; *Silva et al., 2011*). After which the cells are incubated in an appropriate hypotonic solution, to ensure the swelling and bursting of the nuclei (*Moore & Best, 2001*). This is followed by fixation in Carnoy's fixative; preparation of a cell suspension; slide preparation and finally staining (*Moore & Best, 2001*; *Wang et al., 2010*; *Calado et al., 2013*).

Despite the ease of chromosome preparation from eggs and larvae of fish (*Shao et al., 2010*; *Karami et al., 2015*; *Amar-Basulto et al., 2011*; *Botwright, 2015*), cytogenetic studies involving specific tissues of fingerlings, juvenile or adult fish may have some merits over the use of whole larvae or egg. Aside the possibility of having a large amount of tissue from which metaphase chromosome can be isolated multiple times; this method has a pride of place in the characterization of the progenies of intergeneric hybridization (distance crosses) between different fish species from different genus (*Okomoda et al., 2018*). This is because of the need to match phenotypic characteristics of the progenies gotten with their equivalent cytogenetic characters due to the possible presence of ploidy polymorphism (see *Liu et al., 2007*; *Zou et al., 2004*; *Zhao, Zou & Lu, 2015*). However, similarities in the egg and larvae shape of the progenies at this stage of development make it practically very challenging to march phenotypic characters to different ploidy levels.

Chromosome preparation in post larvae fishes can be isolated from different tissues. These includes; fins (*McPhail & Jones, 1966*), gills (*Yoo et al., 2017*), scales (*Ojima, Takayama & Yamamoto, 1972*), kidney (*Zhao, Zou & Lu, 2015*; *Huang et al., 2016*) abdominal cavity fluid (*Fan & Fox, 1990*), gonads (*Tan et al., 2004*) to mention but a few. Despite successfully chromosomes isolation from these tissues, variations in the mitotic cell division rates could result in differences in the quality and quantity of the chromosome observed (*Shelton et al., 1997*). Further, the quantity or amount of tissue used for this processes could affect the concentration of cell suspension and consequently the visibility of chromosome spread. Similarly, the effectiveness of metaphase cellular interaction with cytogenetic chemicals could be affected by the time of "cellular suspension" initiation (e.g., before or after chemical treatments). However, to the knowledge of the researchers, there has been no report in which the tissue amount was optimized, nor the best time for cellular suspension preparation elucidated. Hence, in addition to optimizing colchicine and hypotonic treatment, this study attempted to also optimize different tissue type/ amount as well as identify the appropriate time for making cell suspension and the effects of different aging time on metaphase chromosomes of two important freshwater fishes namely; Asian catfish *Pangasianodon hypophthalmus* (Sauvage, 1878) and African catfish *Clarias gariepinus* (Burchell, 1822).

## MATERIALS AND METHODS

Juveniles of *Pangasianodon hypophthalmus* and *C. gariepinus* (weighing between 10 and 50 g) were obtained from the School of Fisheries and Aquaculture Science hatchery of the Universiti Malaysia Terengganu, in Malaysia. They were acclimatized for 2 weeks in rectangular fiberglass tanks and fed on a commercial diet (35% Crude Protein) until

the experiment was conducted. The method of *Liu et al. (2007)* was used as the basis of protocol optimization in a stepwise manner. For each optimized procedure, five fish were used (per species) for each treatment. Firstly, colchicine concentration and the duration of inoculation were investigated. Juveniles were intramuscularly injected with freshly prepared colchicine solution at 0.01%, 0.025%, and 0.05% colchicine for 1, 3, and 5 h at one ml kg$^{-1}$ of the body weight of the juvenile. Using the selected concentration and duration of colchicine, the suitability of distilled water and KCl solution (0.075M) as a hypotonic solution was tested for 0.5, 1, and 2 h. It is important to state that cytogenetic chemical treatment after colchicine treatment was done in a 1.5 ml tube and a uniform volume of 600 μl of these chemicals was adopted for comparative purposes. Following colchicine and hypotonic solution optimization, 10, 50, 100, and 150 mg of fins, gill, and kidney were extracted and followed by the procedure of *Liu et al. (2007)*.

No alternative fixative, ratio or duration of exposure was considered in this study as freshly prepared Carnoy's solution (methanol-acetic acid at ratio 3:1) is commonly used with a unanimous exposure time of 20 min with three changes (*Fopp-Bayat & Woznicki, 2006*; *Karami et al., 2015*; *Pradeep et al., 2011* etc.). However, the best time for making cell suspension was determined by initiating the process (i.e., chopping the tissue) during hypotonic treatment or prior to slide preparation (after fixation in Carnoy's solution). Before every subsequent step in the formal treatment, the suspension is centrifuged at 2,500 rcf for 10 min, and then the supernatant is discarded leaving one ml of the solution above the cell pellet. The cell is then re-suspended using the next solution. Also, the effect of aging was investigated on the quality of chromosome spread. The cell suspension or tissue was allowed to age for 0, 3, and 7 days in the Carnoy's solution followed by slide preparation. In all trial, slides were prepared by dropping method (two drops of the cell suspension on the slide at one m height) and incubation in 10% Giemsa stain (prepared in 0.01M phosphate buffer at pH 7) for 1 h. Similarly, the metaphase spreads in all the trials were microphotographed using a Nikon Eclipse 80i compound microscope, and the images processed using the NIS element Basic Research software (at 100× magnification).

Chromosome identification and counting were done electronically using the VideoTest Karyo 3.1 (https://karyo-demo.software.informer.com/3.1/). In all the trial, the number of the well-spread chromosome observed was recorded except for the hypotonic treatment where the percentage of the complete/well-spread chromosome was computed. Prior to running analyses of data gotten in this study, normality, and homogeneity of data was tested (*Tabachnick & Fidell, 2001*). For all treatment involving concentration/types vs duration of exposure (i.e., colchicine, and hypotonic solution, respectively), a two-way analysis of variance was employed to evaluate the level of significant differences between each treatment (using Fisher's LSD; $P \leq 0.05$) and their interactions. A similar analysis was done for tissue type vs amount (quantity) and for the time of cell suspension preparation vs aging period. All data analysis in this study was done using Mini tab 14 computer software.

## RESULTS

The result obtained shows that the juveniles injected with 0.05% colchicine (at one ml kg$^{-1}$) for 3 h had better chromosome spread in both species when compared to other

 

concentration and exposure times (Figs. 1A and 2A). Treatment in KCL for 1 h (Figs. 1B and 2B) and the use of 50 mg of gill tissue sample (Figs. 1C and 2C) also proved to be more effective in this study. In addition, preparation of cell suspension before hypotonic treatment and aging for 3 days resulted in well spread chromosomes (Figs. 1D and 2D). A sample of the metaphase chromosome produced in *C. gariepinus* and *Pangasianodon hypophthalmus* are presented in Fig. 3.

## DISCUSSIONS

The choice of a right concentration and duration of exposure of colchicine is very important. This is because insufficient amount could fail to arrest the target cells at metaphase stage (*Rieder & Palazzo, 1992*; *Caperta et al., 2006*), however too high a concentrations or prolonged exposure, on the other hand, may lead to chromosomal condensation (*Wood, Cornwell & Jackson, 2001*). The optimum values recorded in this study for colchicine (i.e., 0.05% for 3 h) is similar to the findings for optimization in post-larval stages of some other fish species (*Liu et al., 2001*; *Botwright, 2015*; *Zhao, Zou & Lu, 2015*; *Huang et al., 2016*). This is, however, at variance with the finding of *Shao et al. (2010)* for eggs and larvae of Japanese flounder (*Paralichthys olivaceus*) and summer flounder (*Paralichthys dentatus*) as they reported best metaphase chromosomes with 0.02% for 1–2 h. While this may be a high enough concentration to penetrate the vitelline membrane of the egg and the tin walls of the larvae, the present study shows that this is not optimum for the post-larvae of African and Asian catfishes. This is despite using intramuscular injection procedure which was thought to be an efficient method to deliver the spindle poison to the fish tissues considering the numerous blood vessels in the skin (*Tan et al., 2004*; *Zhao, Zou & Lu, 2015*; *Huang et al., 2016*).

Subsequent upon mitotic spindle inhibition, it is pivotal to use an appropriate hypotonic solution to swell the nuclei of the mitotic cell to the point of bursting and spread out the chromosomes (*Moore & Best, 2001*). Choosing an improper hypotonic solution and incubation period may result in overlapping or significant loss of chromosomes (*Baksi & Means, 1988*). The efficacy of potassium chloride (KCl 0.075M) over distilled water was demonstrated in this study. In both species, the number of clear metaphase chromosome spreads was significantly higher using the former than the latter. *Karami et al. (2015)* had earlier stated that using KCl caused extensive cell burst and chromosomal loss when compared to distilled water treated larvae of *C. gariepinus*. However, chromosome loss in this study was observed in both hypotonic solutions when the tissue was incubated beyond 1 h, while below this reference point metaphase chromosomes were largely overlapping. The differences in observation of the two studies despite the similarity of the species may be linked to the different developmental stages of fish used.

The present study has for the very first time, shown that too much or too little of tissue amount could affect the number of identifiable metaphase chromosomes. The optimum amount as observed in this study was 50 mg. While the scanty number of chromosome observed below 50 mg could be explained by reduced cell concentration, the observation beyond the reference amount may be because of high cell/tissue concentration which resulted into darkening of the slide background. Hence, this may have covered some well

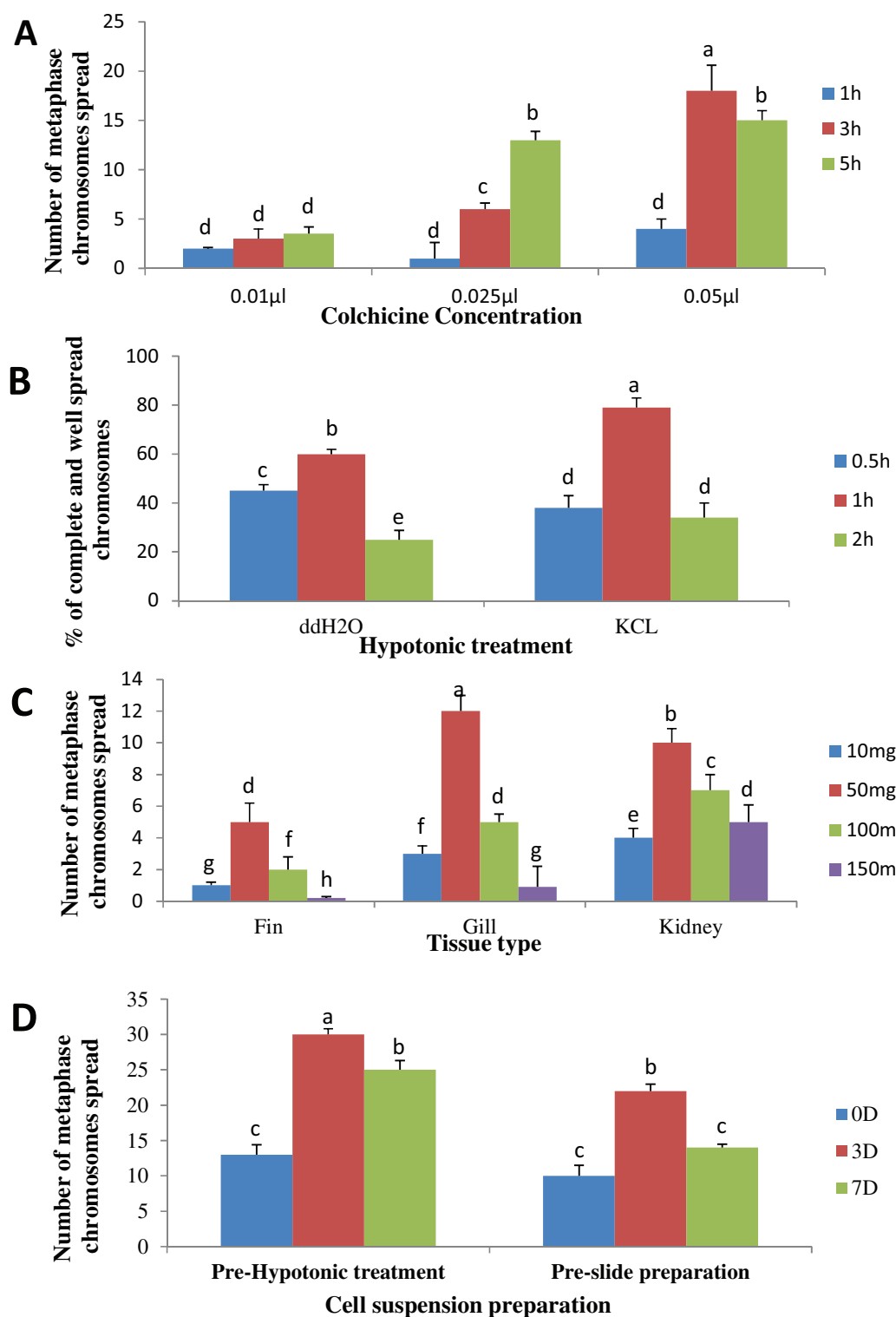

**Figure 1 Effects of different treatments on the number of clear and identifiable metaphase chromosome spreads in *C. gariepinus*.** (A) Colchicine concentration × duration of exposure interaction. (B) Tissues type × mass of tissue interaction. (C) Hypotonic treatment × duration of exposure. (D) Cell suspension preparation × aging time. Data shown are mean ± SE. Bars with different letters are significantly different from each other ($P \leq 0.05$).

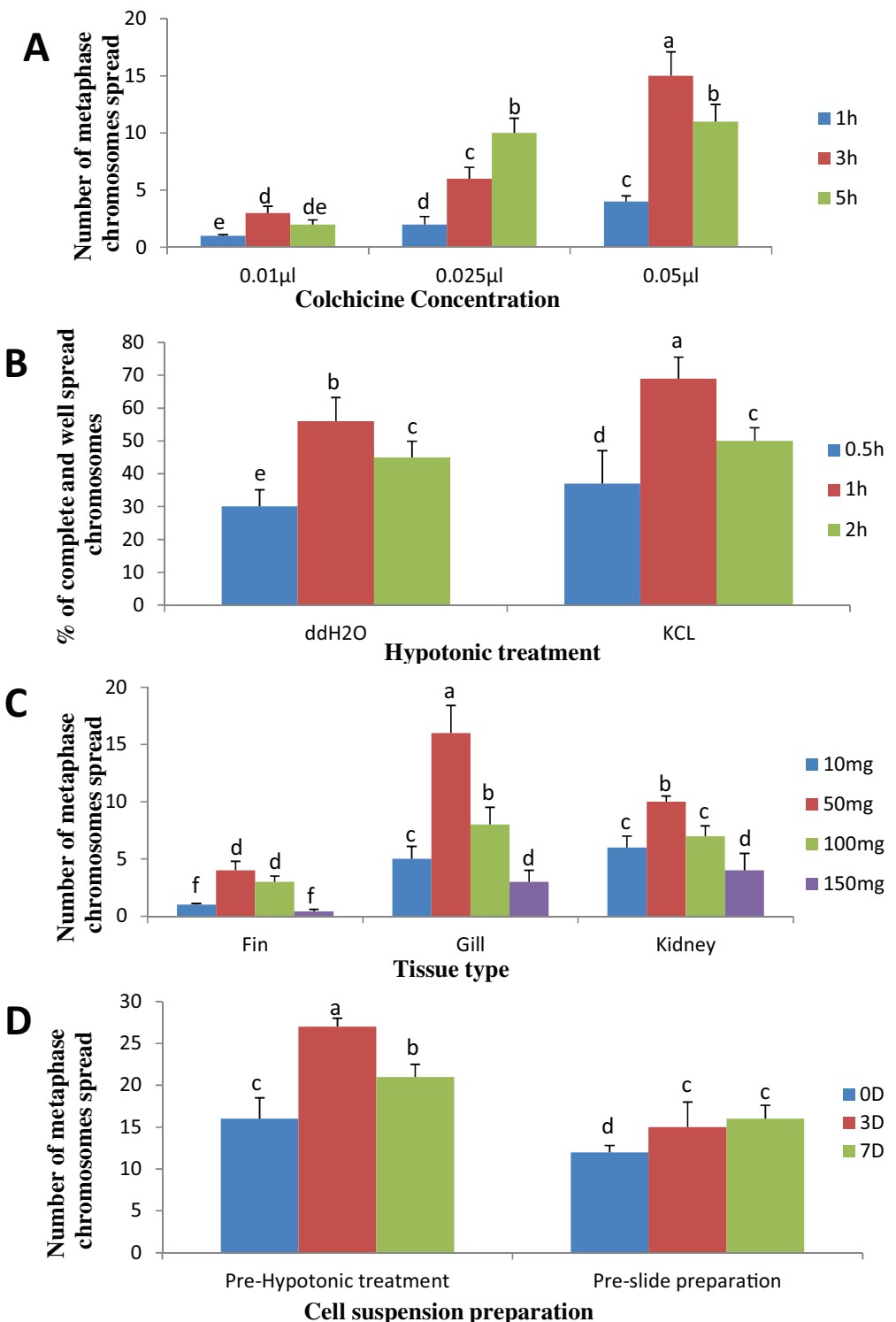

**Figure 2 Effects of different treatments on the number of clear and identifiable metaphase chromosome spreads in *P. hypophthalmus*.** (A) Colchicine concentration × duration of exposure interaction. (B) Tissues type × mass of tissue interaction. (C) Hypotonic treatment × duration of exposure. (D) Cell suspension preparation × aging time. Data shown are mean ± SE. Bars with different letters are significantly different from each other ($P \leq 0.05$).

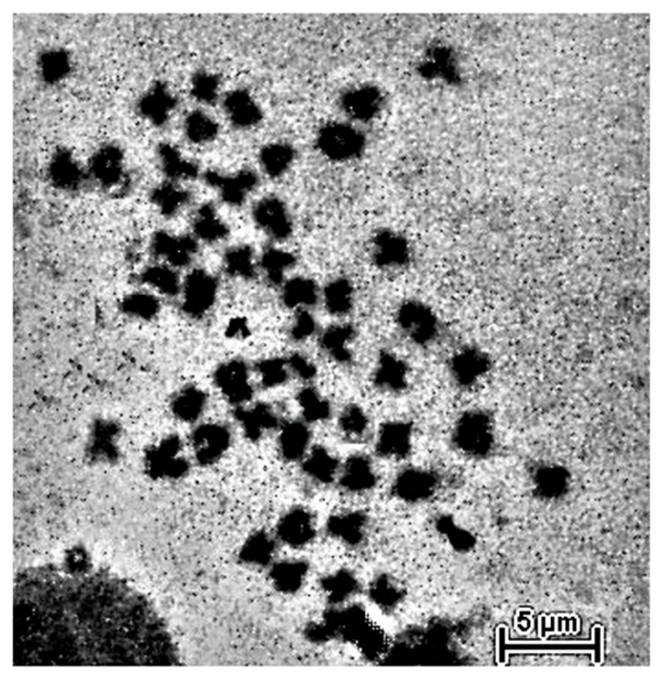

**A**

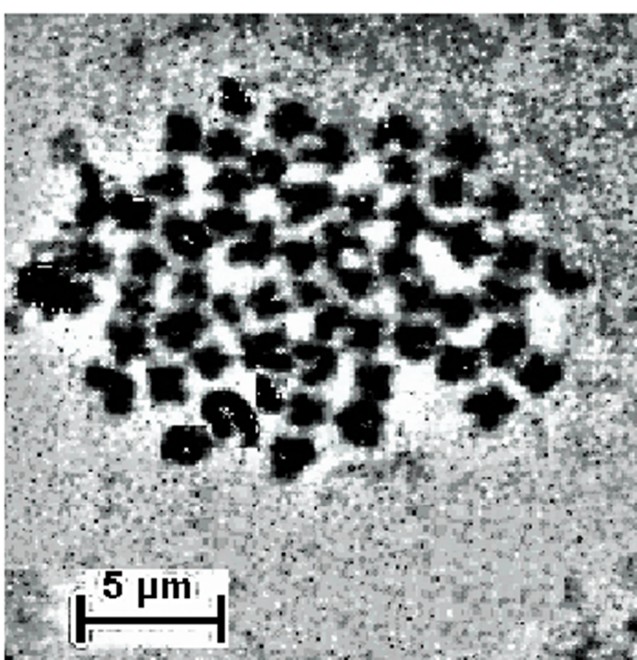

**B**

**Figure 3** Metaphase chromosome of (A) *C. gariepinus* (2*n* = 56) and (B) *P. hypophthalmus* (2*n* = 60). Bar = 5 μm.

spread metaphase chromosome and the spaces between chromatids, consequently making them unobservable. However, this may be remedied (hypothetically) by, respectively, concentrating or diluting the final volume of cell suspension prepared in the Carnoy's solution. The difference in the results from the different tissue is largely connected with the mitotic cell division rates of these tissues as earlier stated. This assumption is similar to the findings of *Shelton et al. (1997)*.

The observation of a higher number of well-spread metaphase chromosomes in the gills may be connected to the vulnerability of the gills to environmental imbalances (*Moyle & Cech, 1996*). This may imply that cell division in this organ could be more rapid to compensate damage from environmental influences. *Wakahara (1972)* had earlier demonstrated that the mitotic cell division rate of the ventral tail-fin epidermis of the larval African clawed frog (*Xenopus laevis*) is largely affected by changes in environmental factors. Therefore, future studies can be designed to understand the effect of environmental changes on the proliferation of the mitotic cell in different tissues of fish. However, the differences in mitotic division rates may also explain the differences in the response to colchicine and hypotonic solutions treatment by the two fishes understudied in this research.

The observation from this study suggests that making cell suspension during hypotonic treatment led to an increased number of observable metaphase chromosome than initiating this process prior to slide preparation. The efficiency of the former is likely connected to the increased surface area of the dissociated cell to the different chemical treatments, hence making their effect more pronounced than the latter whose tissue was intact through the chemical treatment phase until slide preparation. Similarly, aging of the cell suspension or tissue for 3 days in the Carnoy's fixative produces better chromosome spread than preparing slide on the very day of cytogenetic treatment or 7 days after. While the underlining principles responsible for this observation are not well understood, it was speculated that aging might have dissociation effect on the chromosomes. This is because appreciable percentages of the identifiable chromosome in the slide prepared on the very day of cytogenetic treatment were compacted and overlapping, while significant chromosome loss was characteristics of the observations after 7 days of aging. However, these observations are lesser when cells were aged for just 3 days.

## CONCLUSIONS

The metaphase chromosome produced in this study could be used for karyotype analysis. Although species-specific technicalities have been recommended to obtain well-spread metaphase chromosomes in different species (*Karami et al., 2015*), the method described in this study seem to give a satisfactory result for both species using a similar protocol. However, the method of slide preparation by dropping (at one m height) reported in this study could cause loss of chromosome due to technical difficulties. Hence, short distance dropping on the slide with the vapor layer could be an alternative method of choice for researchers adopting this optimized procedure. This protocol may also be effective for cytogenetic studies involving other closely related catfish species such as *C. macrocephalus, Heterobranchus longifilis, Pangasius gigas* etc. This could be the focus of future researches.

## ACKNOWLEDGEMENTS

The authors are indebted to the School of Fisheries and Aquaculture Science, Universiti Malaysia Terengganu, Malaysia for providing juveniles of *P. hypophthalmus* and *C. gariepinus* used in this study. We also acknowledge the help of some technical staffs of the PPSPA hatchery department and laboratory officers of AKUATROP during experimental trials of this study. This study is part of the first author's Ph.D. research.

### Funding

The authors received no funding for this work.

### Competing Interests

The authors declare that they have no competing interests.

### Author Contributions

- Victor T. Okomoda conceived and designed the experiments, performed the experiments, analyzed the data, prepared figures and/or tables, authored or reviewed drafts of the paper, approved the final draft.
- Ivan C.C. Koh conceived and designed the experiments, contributed reagents/materials/analysis tools, authored or reviewed drafts of the paper, approved the final draft.
- Anuar Hassan conceived and designed the experiments, contributed reagents/materials/analysis tools, authored or reviewed drafts of the paper, approved the final draft.
- Thumronk Amornsakun contributed reagents/materials/analysis tools, prepared figures and/or tables, approved the final draft.
- Julia H.Z. Moh performed the experiments, analyzed the data, prepared figures and/or tables, approved the final draft.
- Sheriff Md Shahreza conceived and designed the experiments, analyzed the data, contributed reagents/materials/analysis tools, authored or reviewed drafts of the paper, approved the final draft.

### Data Availability

The raw data are provided in Figure 3 and in the Supplemental Files.

### Supplemental Information

Supplemental information for this article can be found online at http://dx.doi.org/10.7717/peerj.5712#supplemental-information.

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
