# Peer review of "Optimization of the cytogenetic protocol for Pangasianodon hypophthalmus (Sauvage, 1878) and Clarias gariepinus (Burchell, 1822)"

_PeerJ, doi:10.7717/peerj.5712_

## Round 0.1 · original submission · Major Revisions

The reviewers have commented on your above paper. They indicated that it is not acceptable for publication in its present form.
Furthermore, figure 3 is not of good quality and it is vital for this paper. Authors have to provide a better figure of the chromosomes and a demonstration that they are really chromosomes and not artefacts. if you feel that you can suitably address the reviewers' comments (included below), I invite you to revise and resubmit your manuscript.

·

Basic reporting

a. Authors made several unsubstantiated claims .e.g lines 46-49; 52-53 and line 57-59. these generalized claims appears to reflect the authors assumptions. So please clarify.

b. Authors claim ...... 'that the use of intramuscular injections delivered the spindle poison directly into the blood stream' may not be the case as you will need to inject the blood vessels to achieve this assumption (Line 127-129).

c. Based on the sample micrographs presented (i.e figures 3 A and 3 B), the authors assumption that the micrographs represents chromosome spread at metaphase is unproven. the micrographs may just be artifactual changes in the tissues following the treatments.

d. The article appears to have been written by one of the authors only. In my opinion, there is a need to improve on the use of English and this can be attained if the other authors spend some time to assess the article for grammatical errors.
.....see line 77-78 ......(do the authors mean to say "as" or "has" as written?

Experimental design

a. No comment

Validity of the findings

a. In this type of studies, the pictorial evidence (i.e micrographs) are diagnostic (See Purushothaman et al 2016 (DOI 10.7717/peerj.2377); and the statistics presented are derived from the scoring emanating from the data generated from the micrographs. Authors failed to provide a scoring format for their study.

b. Sample micrographs presented (i.e 3A and 3B) are not readable and hence not quantifiable. The reviewer advice the article is not suitable for publication in this journal.

Additional comments

1. Based on the running title, the active word appears to be the word "optimization". So, the article would be better presented with pictorial evidences of the chromosome spread at metaphase showing the various methodologies used in the study.
2. Please provide evidence of how the scoring was made from the micrographs
3. it may be better to label the micrographs presented

Reviewer 2 ·

Basic reporting

Figure 3 is not of good quality. It has a lot of background, it is not possible to see the chromosomes well and the two images are not in the same color scale. I recommend choosing figures that present their results in the best possible way.

Experimental design

Line 95: Some fish chromosome staining protocols use Giemsa 5% for 5 minutes. In the present protocol, you used for 1 hour and in figure 3 we can observe highly colored chromosomes. Would not it be possible to use a coloring for less time, thus improving the preparation time of the material?

The method of preparing slides by dripping 1 meter away can cause the loss of chromosomes, in addition to being difficult technically. Other methods, such as the short-distance dripping on the slide with the vapor layer should be suggested.

Line 86: I could not observe a justification in the introduction for the test of the best time for the preparation of the cellular suspension. The preparation method before hypotonic treatment is accepted and explained in the literature. So, I suggest to justify this test better in the introduction.

Line 104: "..evaluate the interactions.." does not address the objective of ANOVA analysis. The purpose of this analysis could be to verify if there is a difference between treatments or a better explanation.

Validity of the findings

no comment

---

## Round 0.2 · accepted · Accept

I am pleased to confirm that your paper has been accepted for publication. Thank you for submitting your work to this journal.

# Reviewer 2 ·

Basic reporting

no comment

Experimental design

no comment

Validity of the findings

no comment

Additional comments

no comment